



# North African mineral dust sources: new insights from a combined analysis based on 3D dust aerosols distributions, surface winds and ancillary soil parameters

Sophie Vandenbussche[1], Sieglinde Callewaert[1], Kerstin Schepanski[2], and Martine De Mazière[1]

[1]Royal Belgian Institute for Space Aeronomy, 3 avenue circulaire, Brussels, Belgium
[2]Leibniz Institute for Tropospheric Research, Permoserstraße 15 04318 Leipzig, Germany

**Correspondence:** Sophie Vandenbussche (sophie.vandenbussche@aeronomie.be)

**Abstract.** Mineral dust aerosol is a key player in the climate system. Determining dust sources and the spatio-temporal variability of dust emission fluxes is essential to estimating the impact of dust on the atmospheric radiation budget, cloud and precipitation formation processes, the bio-productivity and ultimately the carbon cycle. Although much effort has been put into determining dust sources from satellite observations, geo-locating active dust sources is still challenging and uncertainties in

space and time are evident. One major source of uncertainty is the lack of clear differentiation between near source dust aerosol and transported dust aerosol. In order to reduce this uncertainty, we use 3 dimensional information on the distribution of dust aerosol suspended in the atmosphere calculated from spectral measurements obtained by the Infrared Atmospheric Sounding Interferometer (IASI) by using the Mineral Aerosols Profiling from Infrared Radiance (MAPIR) algorithm. In addition to standard dust products from satellite observations, which provide 2 dimensional information on the horizontal distribution of

dust, MAPIR allows for retrieving additional information on the vertical distribution of dust plumes. This ultimately enables us to separate between near source and transported dust plumes. Combined with information on near-surface wind speed and surface properties, low-altitude dust plumes can be assigned to dust emission events and low-altitude transport regimes can be excluded. Consequently, this technique will reduce the uncertainty in automatically geo-locating active dust sources.

  The findings of our study illustrate the spatio-temporal distribution of North African dust sources based on 9 years of data,

allowing to observe a full seasonal cycle of dust emissions, differentiating morning and afternoon / evening emissions and providing a first glance at long-term changes. In addition, we compare the results of this new method to the results from Schepanski et al. (2012) which manually identify dust sources from Spinning Enhanced Visible and InfraRed Imager Red Green Blue (SEVIRI RGB) images. The comparison illustrates that each method has its strengths and weaknesses that must be taken into account when using the results. This study is of particular importance for understanding future environmental changes due to

a changing climate.





# 1 Introduction

Mineral dust aerosols are one of the most prominent types of tropospheric aerosols accounting for about 35 % of the total aerosol mass with diameter smaller than 10 $\mu m$ (Boucher et al., 2013). Dust aerosols are mainly observed in the so-called "dust belt", a band of northern-hemispheric, subtropical deserts stretching from the Saharan, over the Middle East and Arabia to the Asian deserts. They are mineral particles uplifted from bare soils by sufficiently strong winds. They may stay in suspension for a few days up to even weeks allowing for transport over thousands of kilometres, for example from the Sahara across the Atlantic ocean all the way to the Americas. Dust aerosols are most often found at altitudes below 7 $km$ (Tsamalis et al., 2013) due to their wind-driven emission mechanism and particle size, contrary to mineral ash particles that may be found up to 20 $km$ altitude (Maes et al., 2016).

Mineral dust emission results from wind erosion of bare surfaces and generally refers to the eventual entrainment of soil particles into the atmosphere. Three different mechanisms have been highlighted for these emissions, depending on the particle size and soil composition (e.g., Gherboudj et al., 2016; Marticoréna, 2014): creeping for the largest particles, saltation for the middle size particles (diameter between 70 and 500 $\mu m$) and aerodynamic direct entrainment for the smallest particles. Creeping occurs for particles that cannot be uplifted as they are either too large and/or the wind speeds are too low; they are rolled over the surface and may induce injection of smaller particles upon inelastic impact. Saltation is considered the most efficient emission mechanism (50 to 90 %, Gherboudj et al., 2016) and consists of subsequent short jumps along the surface which induce disaggregation in smaller particles or emission of other particles. This process is also called sand blasting. The third emission mechanism, aerodynamic direct entrainment, occurs when fine particles are directly uplifted by turbulent eddies or strong winds; it is reported to account for only about 1 % of the total dust emissions (Gherboudj et al., 2016).

Dust emission can be considered as a threshold problem: a certain amount of momentum is required to be transferred from the atmosphere onto the soil surface in order to mobilise soil particles and eventually initiate the dust emission process. The momentum required is provided e.g. by horizontal wind, shear stress or turbulence. The amount of momentum needed to initiate particle mobilisation depends on soil surface properties including aerodynamic roughness, particle size distribution, soil type, and soil moisture. Below the so-called threshold for wind erosion (Marticoréna, 2014) or the threshold wind friction velocity (Gherboudj et al., 2016), particles cannot be entrained. That threshold is reported to be in the range of some 5 to 6 $m/s$ (Marsham et al., 2013; Marticoréna, 2014). As the vertical dust emission flux scales non-linearly with near-surface wind speed, most intense emissions occur when the surface wind exceeds 8 to 10 $m/s$ (Kocha et al., 2013; Marticoréna, 2014).

Two major mechanisms are known to be responsible for intense surface winds: the turbulent-induced breakdown of the nocturnal low-level jets (LLJ) and convective events (e.g., Schepanski et al., 2009; Knippertz and Todd, 2012; Heinold et al., 2013; Allen et al., 2013). LLJs are strong horizontal winds forming in the lower troposphere, usually at the top of the nocturnal boundary layer within the residual layer if they form during the night, or at the top of the turbulent and mixed boundary layer if they form during the day. After sunrise, the onset of solar heating causes turbulent mixing resulting in intermittent high surface wind speeds during the mid-morning. This mechanism has been shown to be a significant player for Saharan dust emission, and to have a significant seasonal cycle (e.g., Schepanski et al., 2009; Fiedler et al., 2013). LLJs are frequently present during





the winter over the Bodélé depression and other source areas in the Central and East Sahara, and during the summer over the West Sahara, the Sahel and northern Libya (Fiedler et al., 2013). Convective events driving dust emission are of many different origins and occur at different scales, from the micro scale to the synoptic scale (Knippertz and Todd, 2012; Knippertz, 2014). For example, at the synoptic scale, cyclones form south of the Atlas mountains during spring eventually moving eastward along

the Mediterranean coast (Schepanski et al., 2009; Knippertz and Todd, 2012). A prominent example for meteorological drivers at the mesoscale (up to 500 $m$) are the so-called "haboobs": dust fronts forming due to the down-burst of cold, humid and thus dense air related to deep moist convection that grows during the summer late afternoons and nights (Schepanski et al., 2009; Knippertz and Todd, 2012; Heinold et al., 2013). They occur frequently in the Sahel and southern Sahara (Knippertz, 2014), and can raise substantial walls of dust (Allen et al., 2013). At the very small micro-scale, there are the "dust devils"

(compact rotating dry dust plumes of about 10 $m$ diameter) and larger non-rotating dusty plumes (100 $m$ diameter) occurring at the time scale of some 10 minutes to an hour (Knippertz and Todd, 2012; Knippertz, 2014). They occur under dry conditions and intense surface heating, and may reach 1 to 2 $km$ altitude (Allen et al., 2013). They are extremely difficult to detect from satellite data due to their small scale in both time and space.

Dust emissions are known to follow a diurnal cycle rather than being equally distributed over the day (e.g., Schepanski

et al., 2009; Heinold et al., 2013; Kocha et al., 2013; Banks et al., 2014) with two maxima: a sharp and temporally rather distinct maximum during the local morning hours usually attributed to the turbulence-induced break-down of the LLJ, and a temporally extended maximum during the late afternoon and night, mostly due to convective events (Heinold et al., 2013). Each source area is susceptible to experience either one or both of these mechanisms, with possible seasonal variations regarding predominance linked to wind seasonality (Schepanski et al., 2017).

Although mineral aerosols are in essence natural particles emitted through natural mechanisms, part of their emissions and therefore part of the atmospheric dust burden is induced by human activities. Indeed, these activities disturb the soil and modify the vegetation cover, often leading to increasing surface erodibility potentials. Besides human-induced changes, dust emissions are also indirectly impacted by environmental changes due to climate change in general, which impact the soil temperature and soil moisture, and ultimately the wind patterns (Choobari et al., 2014). Ginoux et al. (2012) used MODIS data to extract

information on mineral dust sources and distinguished between natural and anthropogenic origin of the emissions. They showed that most of the dust sources in the Sahel can be linked to human activities, while most of the sources in the Saharan desert are purely natural.

African dust sources can be studied using in situ measurements, satellite measurements or models. There is a significant variation in the results of different studies found in the literature, linked to the strengths and weaknesses of each measure-

ment system, of each analysis method, and to the local time of the measurements. Schepanski et al. (2012) have compared two different approaches identifying dust sources: (1) via manual plume back-tracking to their point of first appearance by examining 15-minute MSG-SEVIRI InfraRed dust imagery; (2) via relating the frequency of occurrence of increased dust aerosol optical depths (AOD) to dust source activity using daily (noon-time) MODIS Deep Blue AOD and OMI aerosol index (AI). The comparison illustrates significant differences among the two dust source identification techniques, largely due to the

temporal off-set between dust source activation (morning) and satellite overpass (noon). Ashpole and Washington (2013) have





developed a way of automatically tracking dust plumes observed by SEVIRI to identify their source region but only applied it to a small geographical and temporal sample. Parajuli and Yang (2017) have jointly analysed MODIS Deep-Blue AOD with wind and surface fields for the Bodélé area. Parajuli and Zender (2017) connected the dust emission to local geomorphologic features. This is only a sample of the existing dust source analyses, showing a growing interest in the scientific community and

especially new analyses combining different data sets.

The goal of this work is to provide additional information about mineral dust sources and to complete the current knowledge using still another approach. We propose to identify dust sources from a three-dimensional dust aerosol concentration data set retrieved from measurements obtained by the Infrared Atmospheric Sounding Interferometer (IASI) flying aboard the sun-synchronous MetOP satellites by using the Mineral Aerosol Profiling from Infrared Radiances (MAPIR) retrieval algorithm

(Callewaert et al., 2019) to identify low altitude ($< 1\ km$ above ground level) dust events. In a series of analysis steps, we use additional surface wind speed data and surface characteristics to distinguish between transported dust aerosols (at any altitude, also including these at low altitude) and dust aerosols over sources. This eventually provides us with a selection of low-altitude dust plumes which geo-location can be considered as active dust source. We discuss these results for north Africa in terms of dust emission hot spots, seasonal variability, diurnal cycle, and long-term evolution. A special emphasis is given to the

Sahel, which is located at the desert margin – a location that, first, experiences an alternation between dry and wet season, and, second, is sensitive to environmental changes due to a changing climate. Finally, we compare the obtained dust source data set to source locations obtained from the manual backtracking of plumes observed with the 15-minute SEVIRI InfraRed dust index (Schepanski et al., 2007, 2009, 2012).

## 2 Instruments and data

The twice-daily overpass, one overpass during the mid-morning and one during the late afternoon, makes IASI measurements particularly interesting for studying African dust emission. As outlined above, northern African dust emission shows a diurnal cycle with two distinct peaks, one dominating during the morning and a second one during the afternoon/evening (Schepanski et al., 2009, 2017; Heinold et al., 2013). As IASI passes over at 09:30 Local Solar Time (LST), the mid-morning measurements coincide with the mid-morning dust emission peak driven by the down-breaking of the nocturnal LLJs. This process is respon-

sible for a significant part of the dust uplift (Schepanski et al., 2009). The evening measurements at 21:30 LST occur around the time of the second dust emission peak in the afternoon/evening (Heinold et al., 2013). Using IASI therefore allows to gain information at times right after the two diurnal dust emission peaks. IASI therefore has the potential to capture atmospheric dust concentrations close to the actual dust emission event allowing for a more accurate estimate on the amount of dust emitted than instruments measuring at mid-afternoon (13:30 LST) such as MODIS on-board Aqua, CALIOP, or OMI, which observe

mainly dust downwind from sources (Schepanski et al., 2012).

Almost all studies aiming at investigating dust sources undertaken up to now are using satellite total column products and neither distinguish between different aerosol species nor include any vertical information as, e.g., the aerosol layer altitude. In order to distinguish between dust plumes originating from local emissions and these being advected from distant sources, the





criteria based on the Angström exponent are applied. The Angström exponent is a proxy for the particle size which distribution

changes with residence time and transport distance as large particles drop out more quickly (Allen et al., 2013). However, in case of night-time measurements, the Angström exponent criterion is not applicable as the retrieval of the Angström exponent is defined based on solar wavelengths.

On the contrary, IASI's measurements at thermal infrared (TIR) wavelength offer the advantage that the TIR atmospheric window is sensitive only to coarse mode aerosols, including mineral aerosols (desert dust and volcanic ash) and sea spray

aerosols. The latter do not have absorption bands in the TIR, therefore only their scattering effect could be seen, which is found to remain rather low. In addition, sea spray aerosols are not expected above dust source areas either and therefore can be neglected in this study. Mineral aerosols have significant absorption bands in the TIR. Thus, IASI measurements allow for deriving a mineral aerosol product with no influence from other aerosol types possibly present over source areas and thus provide very good prerequisites for assessing dust sources.

In a nutshell, the strengths of IASI data for dust source studies are: (1) IASI provides 10 years of data already and more to come. (2) The entire globe is covered twice per day at very interesting times regarding the monitoring of dust emissions. (3) The data set can be used to retrieve continuously and consistently the vertical distribution of mineral aerosols (see Sect. 2.1).

## 2.1 IASI/MAPIR 3D dust concentration data

The Infrared Atmospheric Sounding Interferometer (IASI) was developed by the Centre National d'Etudes Spatiales (CNES)

and is operated by the European Organisation for the Exploitation of Meteorological Satellites (EUMETSAT). IASI is flying on board the Metop satellite series on a mid-morning sun-synchronous polar orbit (9:30 resp. 21:30 LST equator-crossing). IASI is a Fourier-Transform Michelson interferometer measuring at nadir the Earth and atmosphere emissions and solar backscatter between 645 and 2760 $cm^{-1}$. IASI has a swath width of 2200 $km$ corresponding to a maximum viewing angle of 48.3° on both sides off nadir. Each across-track scan is composed of 30 elementary fields of view, each composed of 4 instantaneous fields of

view of 12 $km$ diameter at sub-satellite point, growing to an ellipse of 39 by 20 $km$ at the edge of the scan line (Clerbaux et al., 2009). IASI provides continuous and consistent data time series developed to match the needs of climate studies. Measurements are available since 2007 and planned until at least 2022 using three successive identical instruments flying on-board Metop-A, B and C. The long-term future is also ensured by the second generation instruments currently in preparation, with higher spectral resolution and better signal-to-noise ratio.

A global data set of vertical profiles of dust concentrations was obtained from IASI cloud-free measurements using the Mineral Aerosol Profiling from Infrared Radiances (MAPIR) algorithm. The MAPIR version 4.1 used in this work is fully described in Callewaert et al. (2019), therefore we only provide a short summary of the main scientific features of the algorithm here. MAPIR operates in 3 retrieval windows in the TIR spectral window: 905 to 927 $cm^{-1}$, 1098 to 1123 $cm^{-1}$ and 1202 to 1204 $cm^{-1}$. All radiative transfer calculations are done with the Radiative Transfer for TOVS (RTTOV) version 12 and are

performed inline for each retrieval. Dust aerosols are parameterised as spheres with a log-normal size distribution of median radius 0.6 $\mu m$, geometric standard deviation of 2, corresponding to an effective radius of 2 $\mu m$. The dust aerosols refractive index (RI) is taken from the Gestion et Etudes des Informations Spectroscopiques Atmosphériques (GEISA, Jacquinet-Husson





et al., 2011) and HIgh-resolution TRANsmission database (HITRAN, Massie, 1994; Massie and Goldman, 2003) dust-like RI based on measurements by Volz (1972, 1973) and Shettle and Fenn (1979).

The MAPIR retrieval is based on the Optimal Estimation Method (OEM, Rodgers, 2000) and iteratively adjusts a state vector, composed of 8 variables: the surface temperature (Ts) and the dust aerosols concentration at the centre of seven 1 $km$ thick layers from from 0.5 to 6.5 $km$ altitude. The a priori dust aerosol vertical profiles are derived from the LIdar climatology of Vertical Aerosol Structure (LIVAS) monthly $1° \times 1°$ climatology, obtained from CALIOP data (Amiridis et al., 2015) from 2007 to 2014.

After the retrieval of mineral dust, a general quality control on the retrieval results tests the goodness of the fit: the root mean square of the spectral residuals (between the measured spectrum and the modelled spectrum after the retrieval) must be lower than 1 $K$. In addition, the total AOD at 10 $μm$ should be at most 5 and the retrieved surface temperature must be between 200 and 350 $K$. The two latter tests are designed to reject cloudy spectra which were not detected by the EUMETSAT IASI cloud product.

**2.2   Dust Source Activation Frequency inferred from MSG-SEVIRI observations**

Meteosat Second Generation (MSG) Spinning Enhanced Visible and InfraRed Imager (SEVIRI) measurements at infrared wavelengths centred around 8.7 $μm$, 10.8 $μm$ and 12.0 $μm$ were used to infer multi-annual information on dust source activity (Schepanski et al., 2007, 2009, 2012). As described by Schepanski et al. (2007), 15-minute dust red-green-blue (RGB) images were analysed manually for identification and individual back tracking of dust plumes to their point of origin, which then was
recorded as initial dust source. By applying this visual back-tracking method, an hourly (2006: 3-hourly) data set on dust source activation frequencies (DSAF) was compiled (Schepanski et al., 2012) covering the four-year period March 2006 to February 2010.

**2.3   ECMWF ERA-5 Surface Winds**

The 10 $m$ wind speed fields used to test whether local atmospheric conditions fostering dust emission were fulfilled were
obtained from the European Centre for Medium-Range Weather Forecasts (ECMWF) ERA-5 reanalysis data, through the Climate Data Store (cds.climate.copernicus.eu). The data are available at hourly time resolution.

**2.4   ESA CCI Land cover and NDVI data**

The land cover and vegetation (Normalised Difference Vegetation INdex, NDVI) data were obtained from the corresponding European Space Agency Climate Change Initiative (ESA CCI) land cover project (http://www.esa-landcover-cci.org). The land
cover data is available globally at 300 $m$ spatial resolution and as 5-years averages for three epochs centred on the years 2000, 2005 and 2010. We used the last one as it covers best the time period of the IASI data. The NDVI climatology describes the natural variability of vegetation cover, snow cover and fire scares (burned areas) at a weekly time resolution and thus provides further constraints on the soil erodibility potential.





## 2.5   ESA CCI Soil moisture data

The surface soil moisture data was obtained from the ESA CCI soil moisture project (http://www.esa-soilmoisture-cci.org) in its version 4.2 (Dorigo et al., 2017; Gruber et al., 2017; Liu et al., 2012). That data set has been generated using active and/or passive microwave space-borne instruments and covers the 36 year period from 1979 to 2016. For this work, we use the data set from the active instruments (on board ERS-1, ERS-2 and Metop-A) because (1) this ensures that the soil moisture was retrieved from measurements at about the same time as the IASI (Metop-A) overpass time, and (2) among the three data sets, the one

from active measurements is the only one expressed in percentage of saturation, which is easier to use in this framework. The other two are expressed in volumetric units $m^3 m^{-3}$. Here, information on soil moisture is used to exclude humid areas as potential dust sources.

## 3   Method

Figure 1 shows a schematic of the combined dust source analysis structuring the present study. Each individual step of this

analysis will be described in the paragraphs hereafter. A hint to the corresponding figures is also given within the schematics. The data at the top of this flow chart (step 1) is the standard MAPIR product with its standard quality control, as described in Sect. 2.1.

### 3.1   Specific filtering of MAPIR data for dust sources studies

For the use of the MAPIR data in dust sources studies, two additional selection criteria are applied in addition to the standard

retrieval quality check: (1) only the data above land surfaces are considered, and (2) only data with sensitivity to the near-surface layer are considered. The term "near-surface layer" in this manuscript refers to the MAPIR 1 $km$ retrieval layer closest to the surface. This additional filtering on near-surface sensitivity is the second step in the schematic shown in Fig. 1.

The sensitivity of the retrieval to any layer can be obtained from the Averaging Kernel (AK) calculated during the OEM retrieval. The AK is a square matrix of the size of the state vector, describing the sensitivity of the retrieval to each parameter

of that state vector (diagonal elements) and the cross-sensitivity between the different state vector parameters (non-diagonal elements). In this study, the need for sensitivity of the retrieval to the near-surface layer is assessed through using the diagonal value of the AK corresponding to that layer (further referred to as $AK_{surface}$). The ideal situation would be $AK_{surface}=1$, meaning that the retrieved concentration in that layer is fully independent from the a priori value. However, in reality, this is very rare and the retrieved concentration depends partly on the a priori and the other layers.

The trace of the AK related to the concentrations, here the sum of 7 values, represents the number of independent pieces of information one can obtain from the retrieval. It is about 2 at best for IASI TIR retrievals (Vandenbussche et al., 2013; Cuesta et al., 2015; Callewaert et al., 2019). Therefore we have set a minimum threshold at 0.25 for $AK_{surface}$. Below that threshold, the sensitivity of the TIR retrieval to the presence of dust aerosols in the near-surface layer is too low and the retrieval, although overall good, does not contain information about that specific layer. Those data are excluded from the source analysis.





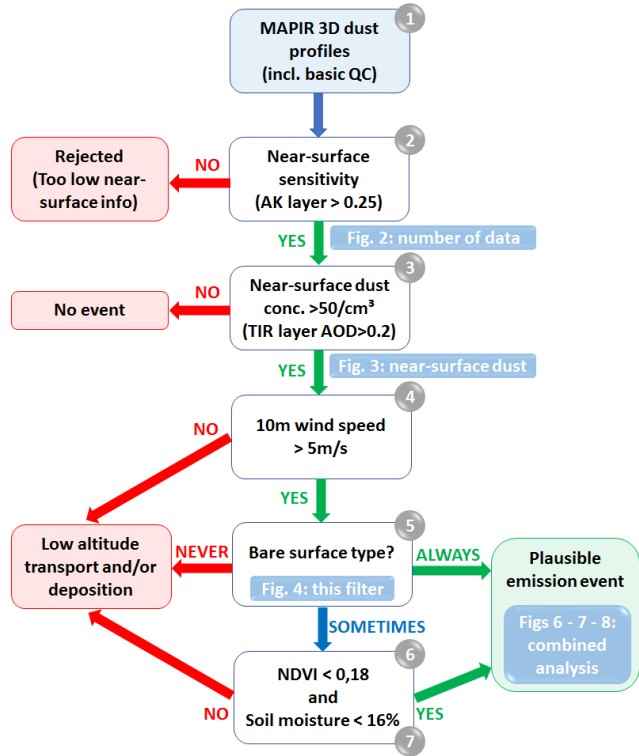

**Figure 1.** Schematic of the combined dust source analysis. Grey numbers indicate the corresponding analysis step numbers, to ease the reference to each step described by the corresponding text paragraphs. Blue boxes refer to the figures illustrating some steps or showing the final results of the combined analysis.

Figure 2 shows the monthly average fraction of IASI retrievals per day passing both the basic quality control (step 1) and specific sensitivity check (step 2) for the entire considered 9-year period 2008-2016. In these averages, each day is given the same weight. White spaces in that figure indicate that no retrieval passes those two filters during the month, showing for example no retrieval sensitivity to the near-surface layer in the central Sahel and south of the Sahel during the summer. This is due to the very low dust AOD in that area during that period, with especially low dust concentration in the near-surface layer,

leading to lesser sensitivity in that layer of the retrieval performed in the logarithmic space, as explained in Callewaert et al. (2019).

## 3.2    Filtering based on MAPIR lowermost layer dust concentration

As explained in Sect. 3.1, the MAPIR retrieved dust concentration in the near-surface layer depends on the real concentration in that layer, but also partly on the a priori assumption. Therefore, the retrieved concentration in a single layer should not be

used quantitatively but still gives evidence for the presence of dust in that layer. To assure robustness, a threshold for MAPIR near-surface layer dust concentration can be defined, above which it is considered reliable. This is step 3 from the method





**Figure 2.** Fraction of days with MAPIR retrievals passing all the selection criteria for the dust source analysis (step 2 of the schematic in Fig. 1), shortened by "available data" in the Figure legend: monthly aggregation of morning and evening data for the years 2008 to 2016.

schematic in Fig. 1. Here, this threshold was set to a concentration of 50 *particles cm$^{-3}$*, or a 10 *μm* AOD of 0.2 (equivalent visible 550 *nm* AOD of 0.35) in the near-surface layer. For comparison, Schepanski et al. (2012) used a visible total column AOD threshold of 0.5 and Ginoux et al. (2012) used a visible total column dust AOD threshold of 0.2 to identify dust sources from MODIS AOD estimates. Our selected near-surface layer concentration threshold is a stronger constraint than those total column commonly used thresholds, but ensures that those are real events.

Considering that the availability of data for this analysis (steps 1 and 2) is in-homogeneously distributed in space and time as shown in Fig. 2, the number of days with near-surface dust detection are divided by the number of days with available data. The result for years 2008 to 2016 is shown in Fig. 3. White areas refer to data unavailable due to low near-surface sensitivity, as in Fig. 2.

**Figure 3.** Fraction of days with near-surface dust in the MAPIR data set (step 3 of the schematic in Fig. 1): monthly aggregation of morning and evening data for the years 2008 to 2016.

### 3.3 Combined analysis

The MAPIR 3D dust data obtained after steps 2 and 3 allow for inferring near-surface dust presence. However, the presence of dust near the surface can have several reasons: (1) it was just emitted there (or in the vicinity), or (2) it was advected at low altitudes, or (3) it belongs to a descending dust layer due to either gravitational settling or turbulent down mixing. In order to filter for dust emission events we apply further criteria: we use ancillary data to evaluate if local conditions foster plausible dust emission and thus support the link between near-surface layer dust and dust emission event. In particular, we consider surface wind speed, land cover, soil moisture and vegetation index. All these parameters help determining if the observed near-surface layer dust events can plausibly be local emission events, but obviously none of them renders the analysis completely certain.





### 3.3.1 Surface winds

We selected a lower threshold wind velocity of 5 *m/s* to possibly foster dust emission. This threshold is on the lower side (Marticoréna, 2014; Marsham et al., 2013; Kocha et al., 2013), to cope with the well-known underestimation of high wind speed in models (e.g. Largeron et al., 2015). Such high wind speeds must occur within the 12 hours prior to the detection of a MAPIR near-surface dust event in order to have this event classified as plausible local emission. This is based on the hypothesis that when an intense emission event occurs, it will be likely to observe it with IASI even a few hours after the emission as

dust remains suspended at low altitude for some time. For example, the afternoon and early evening dust emissions are still observed as low level dust with IASI at 21h30 local time, although the wind at that time has slowed down. The wind filtering represents step 4 in the method schematic in Fig. 1.

### 3.3.2 Surface state

In the framework of this study, we consider all types of bare soil areas, rain-fed or irrigated croplands, grassland, all types of

sparse vegetation ($<15$ %) or shrubland as erodible land type and thus as potential dust source. As the land cover data (Sect. 2.4) is a multi-annual mean, it does not reflect seasonal vegetation changes. Nevertheless, in order to account for seasonal changes in vegetation we apply additional constraints on vegetation and soil moisture for all land cover types but bare soil. A land cover "filter map" containing the following information was built: (1) non-plausible dust source, (2) plausible dust source, and (3) plausible dust source with additional constraints (see Fig. 4). This represents step 5 in the method schematic. A grid

cell is considered to be a plausible dust source if at least 25 % of its surface is of any bare area type. A grid cell is considered to be a plausible dust source with additional constraints if at least 25 % of its surface is made of any of the accepted types. In all other cases, the grid cell is rejected.

Dense vegetation cover is known to absorb the wind momentum and prevent soil erosion and thus dust emissions. The typical NDVI for bare soil areas is about 0.1 while for grassland in Sahel it is about 0.15 from November to July (dry season) and up

to 0.4 during the summer wet season. Parajuli and Yang (2017) have studied the link between dust emissions and NDVI in the Bodélé depression, and concluded that dust mobilisation is fully suppressed when the NDVI exceeds 0.18, which is therefore used here as threshold for plausible dust emission. This is step 6 in the method schematic.

Soil moisture is increasing the inter-particle cohesive forces in the soil and thus increases the amount of energy that is needed to mobilise soil particles and eventually uplift them into the air. Consequently, the threshold wind speed for dust mobilisation

increases with increasing soil moisture. Here, we consider an upper soil moisture threshold of 16 % (Kim and Choi, 2015). This is step 7 in the method schematic.





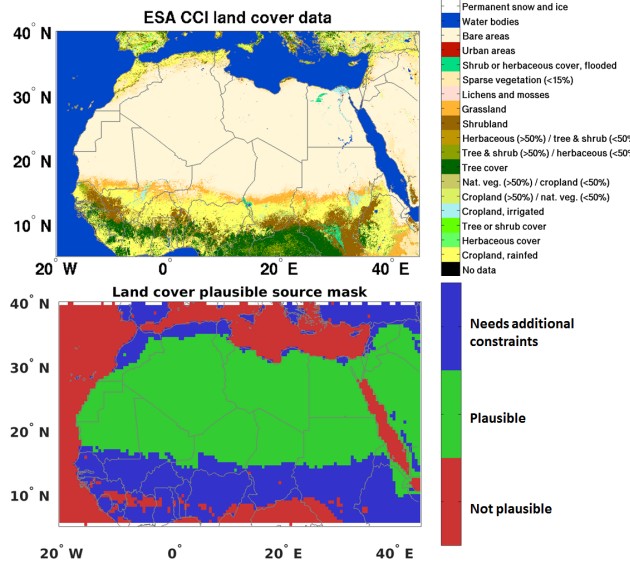

**Figure 4.** Upper plot: ESA CCI land cover data, 2010-centered epoch. Lower plot: corresponding land cover filter for plausible dust sources.

## 4 Results

The resulting retrieved data set on plausible dust source areas covering a 9-year period from 2008 to 2016 will be examined regarding its ability to identify dust source areas, their diurnal activity distribution and their seasonal cycle. Fig. 5 summarises the geographical locations mentioned throughout the discussion.

### 4.1 Saharan dust emission hot spots

Figure 6 shows the monthly all-day results of the combined analysis (corresponding to the final green box in the schematic in Fig. 1) highlighting dust source areas for the years 2008 to 2016. The panel clearly emphasises the seasonal cycles of dust emissions with most frequent emissions occurring during the summer (June, July, August) except for the Bodélé depression area, where dust emission frequency peaks during the winter semester (November to March).

The most striking and predominant hot-spot for dust emission is the Bodélé depression area in central Chad (red label in Fig. 5), south of the Tibesti mountains. It is active throughout the year with a maximum during the winter and a minimum during the summer, which agrees well with the seasonal cycle reported in the literature (Schepanski et al., 2007, 2009, 2012, 2017; Crouvi et al., 2012; Allen et al., 2013). Our analysis highlights further a high fraction of days with plausible dust emission during the winter located west of the Bodélé depression towards the Aïr Mountains in east Niger.

A second major dust emission hot-spot illustrated by our combined analysis is situated in the central Sahara, in the area of the Adrar des Ifhogas and the Aïr Mountains. This area is very active during late spring and summer, especially from June to August, while quiet during the rest of the year, which is consistent with recent literature information (e.g. Schepanski





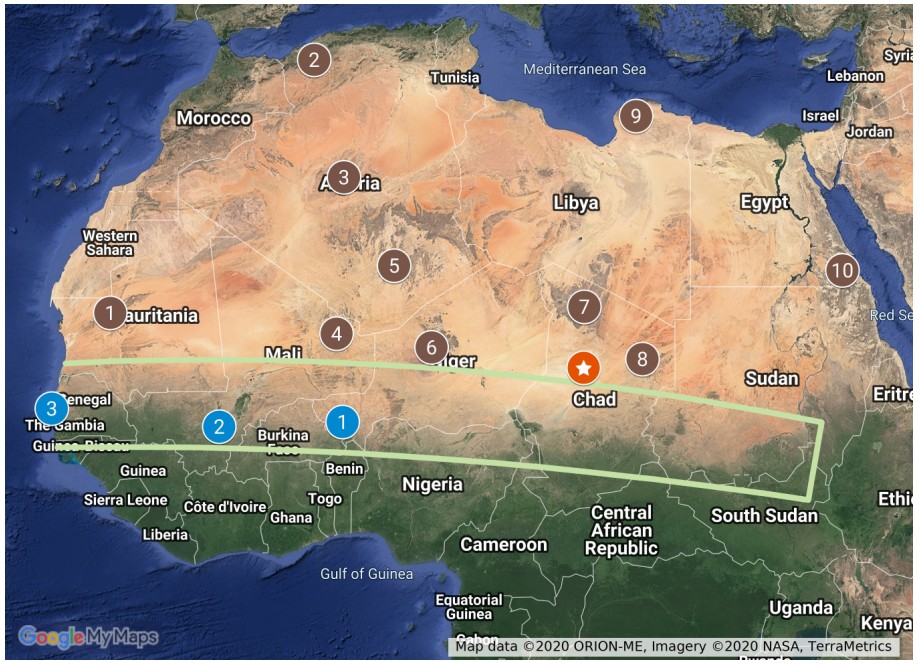

**Figure 5.** Geographical features discussed in the manuscript. The Bodélé depression is shown with a white star in a red circle. The main mountains are shown with numbered brown disks: (1) Addrar Plateau; (2) Saharan Atlas Mountains; (3) Tademaït Plateau; (4) Adrar des Ifhogas; (5) Hoggar Mountains; (6) Aïr Mountains; (7) Tibesti Mountains; (8) Ennedi Mountains; (9) Jabal al-Akhdar and (10) Red Sea Mountains. Ground-based stations discussed in the manuscript are shown with numbered blue disks: (1) M'Bour; (2) Cinzana; (3) Bani-zoumbou. The approximate Sahel boundaries are shown as a green box.

et al., 2007, 2012; Crouvi et al., 2012; Ashpole and Washington, 2013; Evan et al., 2015; Todd and Cavazos-Guerra, 2016;
Schepanski et al., 2017), even though the precise emission locations vary from one study to the other.

Thirdly, a high fraction of days with plausible dust emission is identified during the summer in central Algeria illustrating a dust source region located north of the Hoggar mountains and south-east towards the Libya / Niger / Chad border. It can be characterised as a series of individual small hot-spots. Central Algeria is reported by Ginoux et al. (2012) to be a low occurrences natural source (based on MODIS data). The area close to the Libya / Niger / Chad border is reported to undergo
moderate dust source activation in Schepanski et al. (2007, 2009, 2017).

A fourth dust source area observed is the central-east of Sudan, in the Nile River Basin, with a maximum in July and August, again in agreement with recent literature (Prospero et al., 2002; Schepanski et al., 2007, 2009; Ginoux et al., 2012; Schepanski et al., 2017).

Besides individual hot spot areas, a significant fraction of days with plausible dust emission is observed almost everywhere
in the Sahara during late spring and summer. Smaller, less pronounced hot-spots occur in north-west Algeria (south of the Atlas mountains), in Mauritania and north-west Mali. These areas are also described as dust sources in the literature (Schepanski et al., 2007, 2012; Ashpole and Washington, 2013; Todd and Cavazos-Guerra, 2016; Schepanski et al., 2017).

**Figure 6.** Monthly all-day result of the combined analysis (corresponding to the final green box in the schematic in Fig. 1) for the years 2008 to 2016.

All area highlighted here potentially also include a downwind dust transport component, linked to the transport between sunrise at about 6:00 LST and the measurement time at about 9:30 LST, or the end afternoon convection events also detected a bit later at 21:30 LST. The implicitly included downing transport component results into a shift between actual source and detected plume and thus in a downwind off-set of the assigned source location.

## 4.2 Sahel dust emission hot spots

The Sahel is a known dust source region, mostly discussed with regard to human activities at a sensitive desert margin region (Ginoux et al., 2012; Huang et al., 2015). Klose et al. (2010) showed that dust is mostly present over the Sahel during the winter (December to April) but that it is mainly suspended dust transported from Saharan sources, which accumulates and remains





in suspension due to wind convergence over Sahel on the northern side of the monsoon trough. The whole northern Sahel is highlighted as dust source in Schepanski et al. (2017) although with low DSAF (less than 5 %).

Our method sheds new light on the study of Sahel sources. Indeed, the here presented method by itself intends to isolate local dust emissions from transported dust. In the Sahel, the additional criteria linked to surface conditions are extremely meaningful, allowing to account for the seasonal cycles of vegetation and humidity in addition to the significant changes in winds. What appears in Fig. 6 as a hard cutoff between plausible and not plausible source areas is mostly the result of the applied NDVI and soil moisture filters.

The three most dominant Sahel dust hot spots identified with our method are located in the northern part of the Sahel in the desert margin region: (1) the Mali-Niger border area with a maximum in June-July, (2) southern Mauritania with a maximum in June and July and (3) Mali east of the Mauritanian border also peaking in June and July.

Three ground-based stations were deployed in 2006 in the framework of the African Monsoon Multidisciplinary Analysis (AMMA). This set of stations is called the Sahelian Dust Transect. The stations are found in Banizoumbou (south-west Niger), Cinzana (south Mali) and M'Bour (west Senegal). Their locations are shown in Fig. 5. Using surface PM10 and wind measurements at those stations, Marticoréna et al. (2010) and Kaly et al. (2015) concluded that the enhanced levels of near-surface dust concentrations at those stations during the dry season is mostly due to dust transport from other sources, but some local emissions may contribute towards the end of the dry season (May to July). Bergametti et al. (2017) showed that emission events are rare at Cinzana and Banizoumbou (on less than 2 % of the days) and very short in time (maximum 3 hours), with a maximum during April to July. Two of our identified Sahelian dust hot spots are around two of the Sahelian Dust Transect stations: Cinzana in south Mali and Banizoumbou in south-west Niger, where our analysis also highlights possible local emissions in spring and early summer. Quantitatively, our estimated fraction of days with plausible dust emission is much larger than the stations reported only few percent of dust source activation days.

### 4.3 Diurnal variations

As mentioned in the introduction, dust emissions have a known diurnal cycle and luckily IASI measurements are obtained at interesting times, close to both emission maxima (morning and late afternoon to night). The difference between the morning and evening fraction of days with plausible dust emission may provide information about the main activation mechanism of each emission area and/or period. Figures 7 and 8 show the monthly fraction of days with plausible dust emission respectively for morning (9:30 LST) and evening (21:30 LST) IASI measurements, over the years 2008 to 2016. We remind the reader that in this combined analysis the selection based on the presence of sufficiently high wind speed is done on the 12 hours preceding the IASI measurement. The comparison of Fig. 7 and Fig. 8 lead to three main conclusions. First, the Bodélé depression appears almost exclusively as a morning source, linking it to the LLJ breakup emission mechanism as reported in Fiedler et al. (2013). Secondly, the area to the south-west of the Bodélé depression is almost exclusively observed as afternoon/evening plausible source and might be low altitude transport from the Bodélé emissions. Thirdly, most of other detected major source areas are observed both as morning and afternoon/evening hot-spots, although the afternoon/evening occurrences are lower. Schepanski et al. (2009) also show significantly less dust source activation during the afternoon and evening than during morning.





**Figure 7.** Monthly morning result of the combined analysis (corresponding to the final green box in the schematic in Fig. 1) for the years 2008 to 2016.

### 4.4 Long-term evolution

When analysing the monthly fraction of plausible emission days for each year individually (figures not shown), large inter-annual variability becomes obvious. A robust long-term trend analysis would therefore require longer time series than 9 years. This kind of study will be possible in the future as the IASI mission is currently planned until the mid 2020's, with a scheduled continuation beyond with the successor IASI-NG. A first glance at the temporal evolution of dust source activity is given in Fig. 9, which shows three-year aggregations of our full 9-year analysis: 2008 to 2010, 2011 to 2013 and 2014 to 2016. The three epochs show similar patterns of dust emission hot-spots, although there is a significant variability in the number of plausible emission days. A clear decrease in the frequency of occurrence of dust emission is observed for the central Sahara and Sudan during summer. The activity of the Bodélé depression seems overall stable in the winter and slightly increasing during







**Figure 8.** Monthly evening result of the combined analysis (corresponding to the final green box in the schematic in Fig. 1) for the years 2008 to 2016.

fall. Additional differences are observed between the different time aggregations, clearly showing that comparison between

different dust sources studies should also consider those underlying differences linked to the time period which was analysed. Therefore, robust long-term trends would require longer time series given the high inter-annual variability.

# 5   Comparative Analysis

The results from the new combined dust source detection presented in this manuscript have been compared with those from the SEVIRI manual plume tracking method detailed in Sect. 2.2. These two source studies were undertaken using very different

data (dust concentration vertical profiles from IASI *versus* IR difference composite images from SEVIRI) and source identi-



**Figure 9.** Result of the combined analysis (corresponding to the final green box in the schematic in Fig. 1), seasonal aggregation of all-day data for the years 2008 to 2010 (left column), 2011 to 2013 (central column) and 2014 to 2016 (right column).

fication methods (automatic event selection based on additional data and criteria *versus* manual tracking of each plume to its source area). We will refer to the two methods as respectively the "IASI automated combined analysis" and the "SEVIRI manual tracking". Neither method can be considered to be the "reference truth", the truth probably being somewhere in-between the results of both analyses. The aim of this comparison is to illustrate strengths and weaknesses of both approaches. Only
the dust source patterns obtained with both methods will be compared because a quantitative comparison is meaningless: in the case of the IASI automated combined analysis an event can be observed multiple times (in different grid cells and/or at consecutive IASI time stamps) while in the case of the SEVIRI manual tracking each event is attributed to a single grid cell





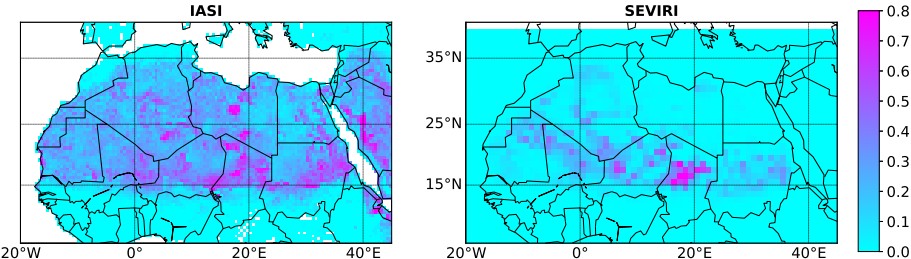

**Figure 10.** Normalised fraction of plausible emission days from the IASI automated combined analysis (left) and normalised dust source activation frequency from SEVIRI manual dust plume tracking (right): annual aggregation of all-day data for the years 2008 to 2009.

and time stamp. Therefore, the results from each analysis were normalised (divided by their respective maximum value) before being compared. These comparisons are shown with a different colour scheme to avoid any possible confusion with the other

figures.

## 5.1 Spatial pattern comparison

There are common source patterns in both analyses: the Bodélé depression, the area between the Hoggar and Tibesti Moutains, west of the Aïr Mountains, east of the Adrar des Ifoghas, the Tademaït Plateau, Jabal al-Akhdar and Sudan. Differences are also observed: the northern Mali and Mauritania and the Saharan Atlas do not stand out from the IASI automated combined

analysis. On the other hand, central Libya and the West African coastal sources do not stand out from the SEVIRI manual tracking analysis. Overall, the IASI automated combined analysis identifies several large areas as "diffuse semi-hot spots".

## 5.2 Diurnal and seasonal cycles

Both analyses agree regarding the most obvious features of the diurnal and seasonal cycles (see Figs. 11 and 12): large intense hot spots during the summer (comprising all highlighted areas in the previous section), Bodélé being active throughout the

year but only in the morning, and an intense summer afternoon / evening source area west of the Aïr and east of the Adrar des Ifoghas Mountains. In this seasonal analysis, the sources in north Mali and Mauritania stand out in the IASI automated combined analysis during the summer, as in the SEVIRI manual tracking analysis. The other differences mentioned in the previous section appear also here. Additionally, Sudan really stands out from the IASI automated combined analysis during the summer afternoons / evenings while being mostly inactive in the SEVIRI manual tracking analysis.

## 390 5.3 Discussion

Most of the differences observed in the two previous sub-sections can be explained as a consequence of the differences between the source identification methods.

A non-negligible part of the differences, where larger source areas are identified from the IASI automated combined analysis (the most striking example is the Sahel area during the summer), originate in the fact that this method counts big events more





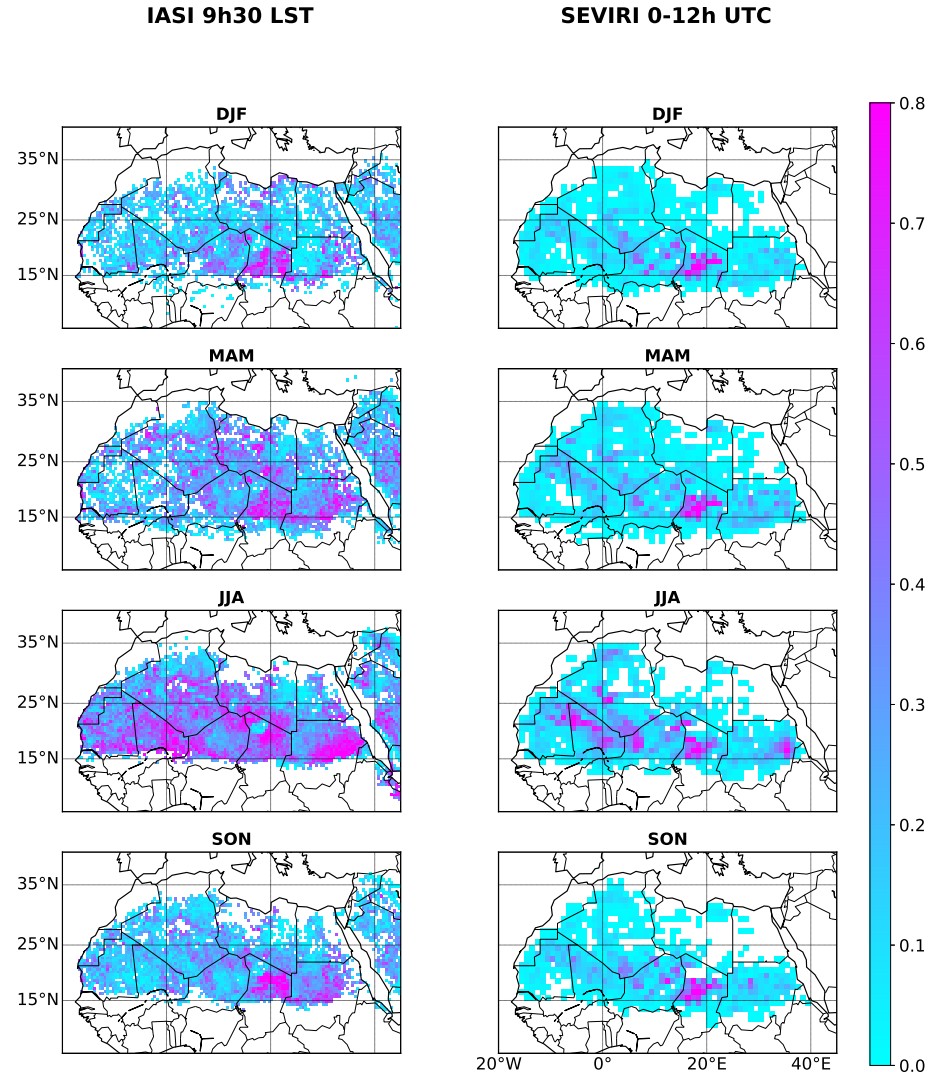

**Figure 11.** Normalised fraction of plausible emission days from the IASI automated combined analysis (left) and normalised dust source activation frequency from SEVIRI manual dust plume tracking (right): seasonal aggregation of day data for the years 2008 to 2009.

than once, identifying each pixel matching plausible emission conditions as a potential source. This most probably leads to an overestimation of the emission occurrences as the low altitude transport after emission is also recorded as a distinct plausible emission event if the local conditions are met. On the other hand, the SEVIRI manual tracking identifies only one pixel and time stamp as source for each event and probably underestimates the continued emissions along the low altitude plume transport if the wind remains strong enough.

The Sudan source area standing out in the evening IASI automated combined analysis is the result of late morning strong winds (after the IASI morning overpass), with as a consequence that it is recorded as evening events while they occur between





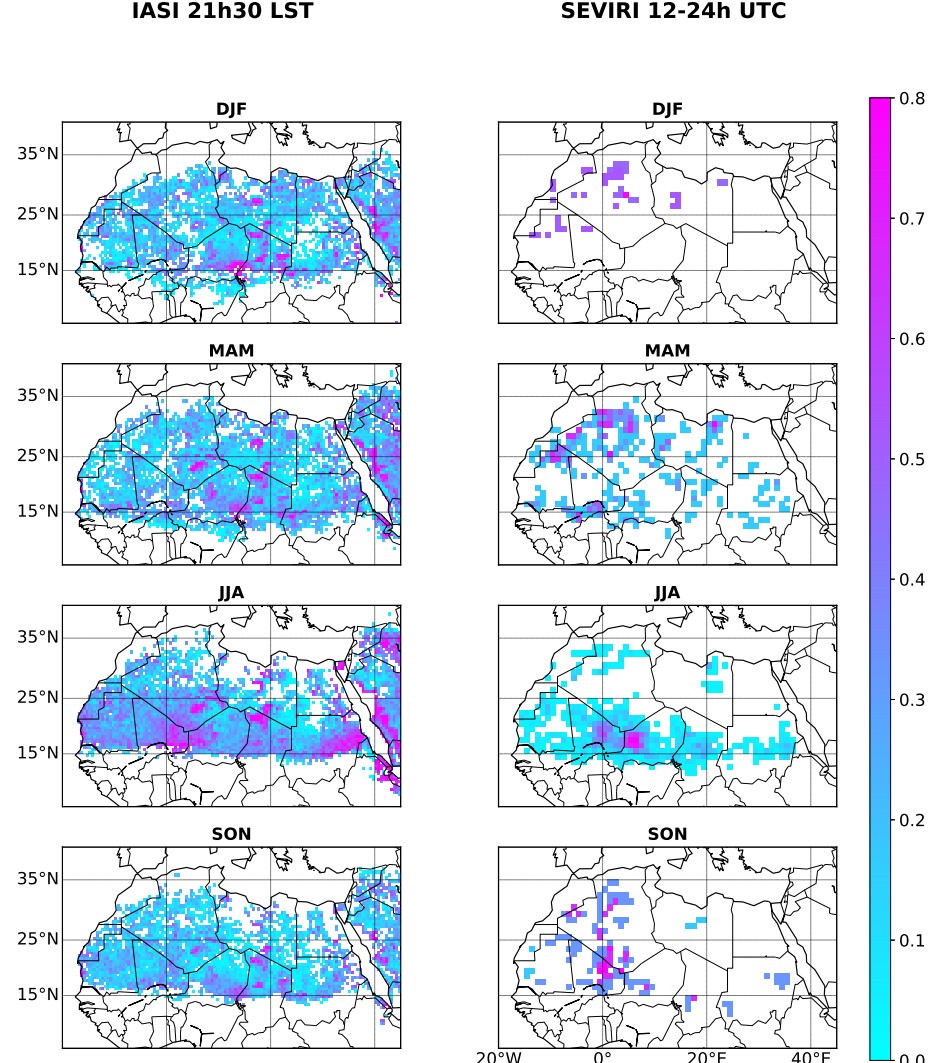

**Figure 12.** Normalised fraction of plausible emission days from the IASI automated combined analysis (left) and normalised dust source activation frequency from SEVIRI manual dust plume tracking (right): seasonal aggregation of night data for the years 2008 to 2009.

9:30 and 12:00 LST. This is supported by two evidences. First, in the IASI automated combined analysis, if the time limit for the existence of high wind speeds is set to 6 hours prior to the overpass time instead of 12 hours, those events are not observed anymore, therefore they are linked to late morning strong winds. Second, in the SEVIRI manual tracking, local emissions are observed in Sudan between 9:00 and 12:00 LST.

The more scattered appearance of the IASI automated combined analysis results is probably linked to the fact that more small plumes are identified in an automated analysis.





The absence of the West African coastal sources from the SEVIRI manual tracking analysis can be linked to the cooler and humid inflow from the ocean, rendering the SEVIRI RGB images much less sensitive to the presence of dust (Banks et al., 2019).

The Saharan Atlas is a place where IASI has very little sensitivity to the near-surface layer, most retrieval results therefore being rejected from the source analysis.

Finally, we do not have an explanation for the central Libya area being highlighted only in the IASI automated combined analysis while this specific spot is not a well-known source area. As the area is identified constantly over time (see Fig. 9), it is either an artefact linked to the local surface properties (infrared emissivity) being misrepresented in the ancillary data used in the MAPIR retrieval, or it is a newly identified dust emission place.

## 6 Conclusions

We describe a new method for the analysis of mineral dust sources, for the first time based on the combined use of a novel 3D dust data set retrieved from IASI measurements, near-surface wind speed data and surface characteristics. In order to identify dust source regions, monthly gridded maps of the fraction of near-surface dusty days are compiled using the quality controlled MAPIR 3D data set. A complex filter based on land cover type, vegetation index, soil moisture and wind speed is applied to best separate plausible emission regions from regions with low-altitude transported dust (including deposition and vertical down mixing of elevated dust layers).

Our analysis highlights and illustrates the added value of using near-surface dust detection instead of total column dust loading. Indeed, based on column dust loading, areas downwind from dust sources may easily appear as hot spots. Our analysis also shows the benefit of using additional information linked to vegetation cover, soil moisture, and wind speed for indicating plausible dust sources. Indeed, the southern Sahel region shows very high occurrences of near-surface dust during the winter, but the surface and wind conditions exclude those places from being plausible dust sources at regional scale. The dust observed there predominantly originates from the Bodélé and the Sahara, which then is transported into the Sahel zone by north-easterly Harmattan winds (e.g. Marticoréna et al., 2010; Kaly et al., 2015).

Our data provide for the first time a monthly aggregation over 9 years, allowing to extend the source analysis to a long period, with a full seasonal pattern. The latter has highlighted the plausibility of significant large-scale dust emissions in central Sahara (with a spring and summer maximum), in the Bodélé depression (with a winter maximum), in central Algeria (summer) and in Sudan (summer). Small-scale dust emission places are also detected all around west and central Sahara during late spring and summer. In the Sahel, three major plausible dust emission places are highlighted during the summer: the Mali-Niger border region, southern Mauritania and Mali east of the Mauritanian border.

The dust sources analysis over north Africa shows a generally good agreement with the literature, demonstrating that our method is indeed suitable for such an analysis. We further compare the results of this new method to the results from Schepanski et al. (2012) which identifies active dust sources from SEVIRI images by manual plume tracking. The two analyses based on very different methods and data overall agree regarding the hot spots and seasonality of dust emissions, but some discrepancies



remain, mainly due to different strengths and weaknesses of each method. This comparison also shows an example of how a dust source analysis depends on the method used to identify those sources. Therefore the use of dust source analyses results must be done with caution and with sufficient knowledge of the method specifics.

In the case of the method proposed here, the main strengths are: (1) dust-only observations twice per day at interesting times,
(2) combining different data in an effort to separate local emissions from transport and (3) the automated processing allowing to quickly analyse large areas for multiple years. The main weaknesses of the method are: (1) the lack of sensitivity of the IASI measurements in some areas for some periods (excluding those from the analysis), (2) the uncertainties linked to all the data sets and thresholds used and (3) the rather limited time resolution of 12

hours. In addition, a specificity of the method is the multiple count of a long-lasting emission event, if the wind and surface
conditions for emission remain along with low altitude dust transport. In that case, probably this new method overestimates the number of events, although it may be argued that if conditions remain stable, emissions do indeed continue along time and along a significant distance and should be counted multiple times.

Building on the interesting capacity of thermal infrared sensors to provide measurements also after sunset, we present a separate analysis of morning and afternoon / evening fraction of plausible dust emission days. This showed reduced occurrences
of plausible emissions during the evening, consistent with the known less frequent dust source activation in the afternoon / evening. Our data also show the Bodélé being an active source only during the morning, consistent with the well-known LLJ emission mechanism in that area.

Future work based on this new automated method will be the characterisation of dust sources in the Middle-East and in Asia, which up to now have been less studied than African dust sources. In the long term, such an analysis will be possible
with longer time series, allowing to study possible trends in dust emissions. The method will be easily adapted to use data from the New Generation IASI, and also most probably data from the InfraRed Sounder on-board Meteosat Third Generation (MTG-IRS). The latter will allow for a better time resolution with a revisit time of maximum 2 hours.

*Data availability.* TEXT

The MAPIR vertical profile data are available upon request to the corresponding author. All ancillary data used in the
combined analysis are freely available as mentioned in Sect. 2.3 to 2.5. The numerical results of the IASI automated combined analysis are available upon request to the corresponding author. The SEVIRI dust source activation frequencies are available upon request to K. Schepanski.

*Author contributions.* S. Vandenbussche developed the initial MAPIR algorithm and supervised the recent improvements, performed the analysis presented in this manuscript and was the main writer. S. Callewaert did the recent improvements of the MAPIR algorithm, in par-
ticular its version 4.1 used in this work. K. Schepanski provided the SEVIRI-based source activation data and participated in the comparison analysis and in the interpretation of all results. M. De Mazière supervised the complete study.



*Competing interests.* There are no competing interests pertaining to this work.

*Acknowledgements.* This work could not have been done without the necessary data sets. We therefore greatly acknowledge the EUMETSAT and EUMETCast service for IASI data used in the MAPIR retrievals, the ESA CCI land cover and soil moisture projects, ECMWF for
producing and Copernicus for distributing the ERA-5 wind fields used in this study. The MAPIR developments were supported by the Belgian Science Policy (Belspo) supplementary researcher program (grants no. WE/35/Q07, R07 and S07), by the European Space Agency (ESA) as part of the Aerosol_cci project phase 2 (grant no. 4000109874/14/I-NB) and by the ESA/Belspo PRODEX IASI.flow phases 2 and 3 (grant no. 4000111402). The specific IASI dust source study was carried out under a programme of, and funded by, the European Space Agency (Living Planet Fellowship entitled "A new method for assessing mineral dust sources using vertical profile information retrieved from IASI
radiances", grant no. 4000116252/16/I-NB). We also acknowledge A.C. Vandaele, N. Kumps, E. De Wachter, V. Letocart, O. Rasson and the BIRA-IASB ICT team for their involvement in the MAPIR scientific developments, the IASI data analysis and the establishment of the MAPIR processing chain. We acknowledge the european COST inDust network (CA16202) for fruitful discussions. Sophie Vandenbussche personally acknowledges S. Plummer for relevant suggestions to successfully lead this analysis.



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
