# Peer review of "North African mineral dust sources: new insights from a combined analysis based on 3D dust aerosols distributions, surface winds and ancillary soil parameters"

_Atmospheric Chemistry and Physics, 2020_

## Referee Comment (RC1) · Anonymous Referee #3 · 21 May 2020

General comments

The analysis gives compelling new evidence on dust emission sources year-round and is an important contribution to the dust source literature for North Africa, with some especially interesting insights on the role of the Sahel. The methodology is well described and the analysis is focused. My main concern relates to the distinction of morning and evening emission events and their link to mesoscale meteorological emission mechanisms, as well as the use of reanalysis winds. One more minor comment is a suggestion to improve the introduction. See below for all comments.

[Figure]

Major comments

1. The results have some interesting implications in the debate on the role of meteorological emission mechanisms and in this paper it's suggested that morning (09:30) dust is linked to LLJ breakdown. I feel if this is link is going to be explicitly made in the paper, the following questions arise in the context of summertime: a. 09:30 is close to the timing of LLJ emission, but during summer isn't it plausible that most observable dust at this time is still due to cold pool outflow (CPO) activity from the night before, especially if the CBL hasn't developed to mix it out of the bottom 1km? See Allen et al. (2013) for issues with using emission timing to infer emission mechanisms. At a glance, SEVIRI also shows much more dust at 09:00 compared to 21:00, but based on Caton Harrison et al. (2019) this appears to be mostly residual CPO dust. b. Can these results be reconciled with evidence from the literature that CPOs are the primary driver of dust emission during summer?

2. I am not aware of any evidence that ERA5 is capable of representing winds associated with CPO activity. Could this algorithm risk unfairly excluding many of those events with high CPO winds that are severely underrepresented in ERA5?

3. For the introduction: please consider discussing more the challenges and opportunities associated with satellite dust source detection/analysis? Maybe less space could be devoted e.g. to the microscale details of dust emission to accommodate this. I feel it would be beneficial to explicitly justify the current research in the context of the literature, other than just providing 'still another approach'. How has dust source detection been accomplished and what exactly is missing that this work addresses? If it's the vertical component, please also compare with previous attempts at this, e.g. Todd and Cavazos-Guerra (2016). See also Bakker et al. (2019) for a recent dust source analysis in ACP. A preprint article by Chédin et al. (2019) on IASI dust observations (https://www.essoar.org/doi/10.1002/essoar.10501119.1) may also be of interest.

4. Surface and soil moisture thresholds (algorithm steps 5 and 6): is there any way you

could give a sense of what proportion of pixels which would have otherwise passed were filtered out by this step? It looks from Figure 4 like mainly the Sahel region will be affected. As you identify later this is a crucial factor as our understanding of Sahel dust sources is limited.

5. Figure 5: The Tademait Plateau itself seems quite unlikely to be a dust source, compared to the large array of sebkhas in the Tidihelt basin directly south (e.g. see Ashpole and Washington, 2013) – please double check whether the plateau is actually collocated with your results?

Technical corrections

A few small grammatical/language issues need addressing but don't affect general readability. Some examples are "estimate on" rather than "estimate of", sentence starting "The Angstrom exponent..." on L124, L191 "that data set" instead of "this data set", L308 "All area"...One repeated error is the use of "allows to" or "allowing to".

L25: "Saharan"

L90: The Ashpole and Washington scheme temporal sample is 2004-2010, which is long but only applied for summer, and the geographical area is the western Sahara which is not complete but does not seem small?

L103: "This eventually..." sentence unclear, please rephrase

L105 ", and," remove first comma

L116 "allows to gain" rephrase

L118: "estimate on" -> "estimate of"

L128 "by contrast" more appropriate than "on the contrary"

L134 "provide very good prerequisites" unclear – perhaps a "good basis"?

L135 Minor but the first point is stated here for the first time whereas 'in a nutshell'

implies it isn't.

Paragraph 1 of section 2.1 would make more sense at the start of section 2

L187 "scares" -> "scars"

L200 "a hint to..." I'm not sure what this means. Perhaps you mean that the corresponding figure is referenced in the schematics?

Figure 2: a discrete colour bar would be much easier to read than a continuous one. The same applies to all other map figures.

L236 Not clear what "those" refers to

Section 4.1 please refer to the label numbers in Fig 5 as you discuss geographical locations.

Section 5.1 please continue referring to label numbers in Fig 5 where possible.

L402 should be "two pieces of evidence"

L451 "emissions do..." unclear, please rephrase
* * *

---

## Referee Comment (RC2) · Anonymous Referee #2 · 2 Jun 2020

This manuscript described a new algorithm for applying satellite product for identifying dust particles at source and downwind areas, with solid method and detailed analysis of employed dataset. As mentioned in the manuscript, dust plays an important role in the climate system and this study would be a good supplement to constrain the emission, transport, and atmospheric burden of dust in addition to the modeling method and other satellite products. The manuscript is well organized with fluent writing, therefore I would like to recommend it to be accepted for publication with minor revision if the following comments could be properly addressed:

[Figure]

Comment#1. Line#82-95: what's the limitations/advantages of these approaches, and how the new method promoted in this study can address those challenges?

Comment#2. Line#75: why anthropogenic influence is mentioned here? Can the method described in this study be applied for human-driven dust from cropland road dust?

Comment#3. Line#96: "The goal of this work . . . " this sentence doesn't give any clear information with "additional information" and "still another approach"

Comment#4. Line#121&Line#130: It's interesting to comment on "aerosol species", are there any satellite products addition to TIR available to distinguish mineral dust particle from other aerosols? How sea salt is excluded from TIR? Can other satellite sensors with TIR bands be used with the same method to identify dust as IASI?

Comment#5. Line#163: So MAPIR relies on CLIPSO to determine vertical profile of dust?

Comment#6. Line#168: this is confusing, line#150 mentioned the data is from IASI cloud-free measurements, so the AOD=5 and 200-350K is setup as double screening of cloud spectra?

Comment#7. Line#190: what's the depth of soil moisture data, and what's the depth we need to pay attention for considering dust elevation?

Comment#8. Line#250-252: models use "friction velocity", it is not equivalent to the wind velocity. Have you tried sensitivity analysis by changing the value of this threshold?

Comment#9. Line#269: As the land cover is multi-annual mean, what's the temporal resolution of NDVI?

Comment#10. Figure 5: I would rather prefer not use google maps figure in a scientific journal publication, there are a lot dataset could be used here to identify the

geographical condition.

Comment#11. Figure7&Figure8: It's interesting to see the diurnal pattern, please discuss why some places and months (e.g., Aug) show larger difference.

Comment#12. Section4.4: long-term trend is of special importance to understand the climate effect of dust, please consider add time-serious analysis of the result, for example, annual variations for the whole study period, instead of 3-year aggregation.

---

## Author Comment (AC1) · 17 Jul 2020

**Author comment on "North African mineral dust sources: new insights from a combined analysis based on 3D dust aerosols distributions, surface winds and ancillary soil parameters"**

Sophie Vandenbussche      Sieglinde Callewaert      Kerstin Schepanski

Martine De Mazière

July 16, 2020

We thank the 2 referees for the time they spent on the manuscript, and their constructive comments and suggestions about our manuscript, which we all have considered carefully. Here under we have copied their comments in bold, and our answers follow in regular font. Changes made to the manuscript are highlighted here by quotation: omitted / removed parts are crossed out; added parts are highlighted in bold. All line numbers refer to the ACPD first manuscript.

**Referee 2 - Referee comment 2**

**This manuscript described a new algorithm for applying satellite product for identifying dust particles at source and downwind areas, with solid method and detailed analysis of employed dataset. As mentioned in the manuscript, dust plays an important role in the climate system and this study would be a good supplement to constrain the emission, transport, and atmospheric burden of dust in addition to the modeling method and other satellite products. The manuscript is well organized with fluent writing, therefore I would like to recommend it to be accepted for publication with minor revision if the following comments could be properly addressed:**

**Comment#1. Line#82-95: what's the limitations / advantages of these approaches, and how the new method promoted in this study can address those challenges?**

Table 1 here under summarises briefly the limitations and advantages of the various approaches discussed in the manuscript lines 82-95. Following comments from the other referee, we added the work of Todd and Cavazos-Guerrera (2016) and Caton Harrisson et al. (2019). For completion, we also included Ginoux et al. (2012). The table is now included in the revised manuscript. The manuscript paragraph was modified as follows (including changes linked to comments from referee # 3):

> African dust sources can be studied using in situ measurements, satellite measurements or models. There is a significant variation in the results of different studies found in the literature, linked to the strengths and weaknesses of each measurement system, of each analysis method, and to the local time of the measurements. Schepanski et al. (2012) have compared two different approaches identifying dust sources: (1) via manual plume back-tracking to their point of first appearance by examining 15-minute MSG-SEVIRI InfraRed dust imagery; (2) via relating the frequency of occurrence of increased dust aerosol optical depths (AOD) to dust source activity using daily (noon-time) MODIS Deep Blue AOD and OMI aerosol index (AI). The comparison illustrates significant differences among the two dust source identification techniques, largely due to the temporal off-set between dust source activation (morning) and satellite overpass (noon). **Ginoux et al. (2012) used MODIS Deep-Blue data to retrieve dust sources and attributed them to anthropogenic or natural origin with the help of land use and hydrology data.** Ashpole and Washington (2013) have developed a way of automatically tracking dust plumes observed by SE-VIRI to identify their source region  **and applied it to summertime western Sahara. Recently, Caton Harrison et al. (2019) improved that method and extended the time period analysed, still focusing on the summertime western Sahara. Todd and Cavazos-Guerra (2016) used CALIOP aerosol**

**vertical profiles to calculate a dust emission index for the Sahara during summer.** Parajuli and Yang (2017) have jointly analysed MODIS Deep-Blue AOD with wind and surface fields for the Bodélé area. Parajuli and Zender (2017) **further** connected the dust emission to local geomorphologic features. **The limitations and advantages of those approaches are briefly summarised in Table 1.** This is only a **sub**sample of the existing dust source analyses **based on satellite measurements**, showing a growing interest in the scientific community  **for** especially new analyses combining different data sets.

| Approach | Limitations | Advantages |
| --- | --- | --- |
| Schepanski et al. (2012) (SEVIRI) | manual time consuming | 15 min temporal resolution back-tracking of individual plumes day and night |
| Schepanski et al. (2012) (MODIS / OMI) | temporal offset wrt emission once per day total column all aerosols | automatic long time series |
| Ginoux et al. (2012) (added in the revised manuscript) | temporal offset wrt emission once per day total column | natural vs anthropo. source attribution global dust sel. based on size and opt. prop. |
| Ashpole and Washington (2013), Caton Harrison et al. (2019) | spatial and temporal constraints | automatic 15 min temporal resolution |
| Todd and Cavazos-Guerra (2016) (added in the revised manuscript) | temporal offset wrt emission only summer time poor geographic coverage | automatic high resolution vertical profile |
| Parajuli and Yang (2017) | same as Schepanski et al. MODIS limited to Bodélé | automatic using wind and surface parameters |
| Parajuli and Zender (2017) | only applied to Bodélé same as Schepanski et al. MODIS | generic approach AOD and wind speed correlation |

Table 1: Brief summary of limitation and advantages of a selection of approaches using satellite data to analyse dust sources.

Our approach addresses some limitations from the previous approaches by providing:

- An automatic approach

- Measurements twice per day at times closer after the expected emission peaks than with MODIS

- An aerosol data set with specificity towards mineral aerosols

- Information on the vertical distribution of the aerosols

- A combined approach using additional parameters: wind speed, surface state

The first sentence line 96 has been removed (see also answer to comment 3) and the following paragraph was slightly rephrased to include those specific considerations:

>  **Here, we** propose **a novel approach** to **automatically** identify dust sources from a **twice daily** three-dimensional dust aerosol concentration data set retrieved from measurements obtained by the Infrared Atmospheric Sounding Interferometer (IASI) flying aboard the sun- synchronous MetOP satellites by using the Mineral Aerosol Profiling from Infrared Radiances (MAPIR) retrieval algorithm (Callewaert et al., 2019) to identify low altitude (< 1 km above ground level) dust events. **These measurements are acquired at 9h30 and 21h30 local solar time, relatively closely after the expected dust emission peaks.** In a series of analysis steps, we use additional surface wind speed data and surface characteristics to distinguish between transported dust aerosols (at any altitude, also including these at low altitude) and dust aerosols over sources.  **At the end of the process we obtain** a selection of low-altitude dust plumes which geo-location can be considered as active dust source. [...]

In addition, although this study shows the algorithm's feasibility for North Africa, it can be applied globally. The same algorithm can be applied for future satellite missions with thermal infrared sensors such as IASI-NG and most probably IRS. Both these future possibilities are mentioned in the Conclusion section of the manuscript.

**Comment#2. Line#75: why anthropogenic influence is mentioned here? Can the method described in this study be applied for human-driven dust from cropland road dust?**

In this paragraph, we mostly refer to the human induced land use and environmental changes due to climate changes which impact the spatial extent and susceptibility of desert dust sources to wind erosion. This includes dust from croplands .

As far as the road dust is concerned, we are not aware of any thermal infrared satellite based analysis. Those events are much smaller in space and thus difficult to detect from space.

**Comment#3. Line#96: "The goal of this work . . . " this sentence doesn't give any clear information with "additional information" and "still another approach"**

The details of the new approach are given in the next lines. Indeed that first introductory sentence does not say much and is now removed. The paragraph was modified according to this comment and Comment 1, and its new version is found in the answer to Comment 1.

**Comment#4. Line#121&Line#130: It's interesting to comment on "aerosol species", are there any satellite products addition to TIR available to distinguish mineral dust particle from other aerosols? How sea salt is excluded from TIR? Can other satellite sensors with TIR bands be used with the same method to identify dust as IASI?**

There are, to our knowledge, two satellite remote sensing techniques with some selectivity regarding aerosols. The first is the use of TIR as done in this manuscript, with aerosol sensitivity almost exclusively to mineral aerosols (mineral dust and volcanic ash). The second is the UV aerosol absorbing index which allows to distinguish UV absorbing aerosols (mineral aerosols and biomass burning) from non-absorbing aerosols. If mineral dust is the specific target, the TIR based retrievals are the best option.

Sea salt is not excluded from TIR (coarse particles) but at this point there is no evidence that it has a significant impact on the satellite measured radiance and we did not observe it in our MAPIR ocean retrievals. Ensuring the absence of impact of sea salt would require a specific study, which is out of the scope of this manuscript as, in any case, sea salt aerosols are not expected at desert dust source areas.

Other satellite TIR sensors than IASI could be used to observe mineral dust. Their sensitivity to dust would however depend on their spectral resolution and signal-to-noise ratio. For example, we plan to test the adaptation of our MAPIR retrieval to the future IRS instrument on board MTG-S platforms.

**Comment#5. Line#163: So MAPIR relies on CLIPSO to determine vertical profile of dust?**

MAPIR is an optimal estimation retrieval. As such it requires an a priori knowledge of the parameters to be retrieved, used as constraint and starting point for the retrieval. That a priori is usually a climatology, with mean and variability. In our case, the mean value for the a priori is a climatology based on CALIOP. We use a monthly mean along 8 years of CALIOP data. The variability is set to 100%. This is detailed in the algorithm description paper (doi:10.5194/amt-12-3673-2019). So, in short, we do rely on a monthly mean along 8 years of CALIOP data as constraint and starting point for the retrieval, but each single MAPIR retrieval provides a distinct vertical profile not at all linked to the single CALIOP measurement occurring closely in time and space.

**Comment#6. Line#168: this is confusing, line#150 mentioned the data is from IASI cloud-free measurements, so the AOD=5 and 200-350K is setup as double screening of cloud spectra?**

Exactly. As mentioned in the manuscript line 168, some clouds are not detected by the EUMETSAT filter and this final test is meant to remove those.

**Comment#7. Line#190: what's the depth of soil moisture data, and what's the depth we need to pay attention for considering dust elevation?**

The soil moisture information is retrieved from remote sensing (not soil sampling). In Dorigo et al, 2017 (the reference publication for the data used) it is mentioned that (section 2.1) "The microwave domain is particularly

useful for the observation of moisture conditions in the upper few centimetres of the soil (Ulaby et al., 1982)."
We modified the sentence line 190 as follows (addition in bold):

> The surface **(upper few centimetres)** soil moisture data was obtained from the ESA CCI soil moisture project ...

For dust emission, only the particles closest to the surface are involved in particle mobilisation processes.

**Comment#8. Line#250-252: models use "friction velocity", it is not equivalent to the wind velocity. Have you tried sensitivity analysis by changing the value of this threshold?**

We agree that dust production models generally use the wind friction velocity instead of the near-surface wind speed in order to estimate the dust emission flux. However, here we do not intend to model dust emission fluxes. Rather, the wind speed data is used to verify that plausible emission conditions were present at the time when and where airborne dust is observed in high concentrations at levels close to the surface. The use of wind data is further to attribute the observed dust event to either a local emission event or transported dust. The event itself is observed separately and independently from the occurring wind condition.
We ran a sensitivity study with a modified wind speed threshold (4 and 6 m/s, respectively). The absolute fraction of days with plausible dust emissions varies (higher for a lower wind speed threshold) but only for the dust emission hot spots.

**Comment#9. Line#269: As the land cover is multi-annual mean, what's the temporal resolution of NDVI?**

NDVI is a weekly data set as described in section 2.4

**Comment#10. Figure 5: I would rather prefer not use google maps figure in a scientific journal publication, there are a lot dataset could be used here to identify the geographical condition.**

We think that this map shows what we intend to show, and google map is not uncommon to see in scientific publications. The imagery itself comes from NASA as clearly indicated on the picture.

**Comment#11. Figure7&Figure8: It's interesting to see the diurnal pattern, please discuss why some places and months (e.g., Aug) show larger difference.**

The difference in diurnal pattern can be explained by the seasonally changing predominance of meteorological drivers fostering dust emission. In particular, the fractional contribution by LLJ versus MCS (haboobs) changes with the seasonally changing atmospheric circulation pattern. These drivers and their seasonal variability are discussed in section 4.3, starting line 344. Also see answer to Referee #3, "Major Comment 1".

**Comment#12. Section4.4: long-term trend is of special importance to understand the climate effect of dust, please consider add time-serious analysis of the result, for example, annual variations for the whole study period, instead of 3-year aggregation.**

We understand that long-term trends are important. We however do mention that more data would be required to be able to address this with any statistical significance and robustness. The year-to-year variability is pretty high and showing annual variations for the whole study period would not, at this point, allow for a statistical regression to determine a trend. Therefore, we decided to only provide a glimpse of how the dust source activity changes along the available time period by comparing three-year averages. Nevertheless, the analysis you are asking for stays in our minds for the future - but additional years of measurements would certainly be needed.

**Referee 3 - Referee Comment 1**

**General comments**

The analysis gives compelling new evidence on dust emission sources year-round and is an important contribution to the dust source literature for North Africa, with some especially interesting insights on the role of the Sahel. The methodology is well described and the analysis is focused. My main concern relates to the distinction of morning and evening emission events and their link to mesoscale meteorological emission mechanisms, as well as the use of reanalysis winds. One more minor comment is a suggestion to improve the introduction. See below for all comments.

**Major comments**

**1. The results have some interesting implications in the debate on the role of meteorological emission mechanisms and in this paper it's suggested that morning (09:30) dust is linked to LLJ breakdown. I feel if this is link is going to be explicitly made in the paper, the following questions arise in the context of summertime: a. 09:30 is close to the timing of LLJ emission, but during summer isn't it plausible that most observable dust at this time is still due to cold pool outflow (CPO) activity from the night before, especially if the CBL hasn't developed to mix it out of the bottom 1km? See Allen et al. (2013) for issues with using emission timing to infer emission mechanisms. At a glance, SEVIRI also shows much more dust at 09:00 compared to 21:00, but based on Caton Harrison et al. (2019) this appears to be mostly residual CPO dust. b. Can these results be reconciled with evidence from the literature that CPOs are the primary driver of dust emission during summer?**

Indeed in the manuscript we suggest that the fact that events are detected in the morning or evening can help link them to different emission mechanisms. Obviously, in the framework of an automatic identification retrieval as presented here, this link cannot be done on an individual case basis and manual inspection as it is done in Allen et al. (2013). So, here, the attribution relies on the estimated emission timing only. It is of course often the case that the dust observed at the time of the satellite overpass may not be locally produced. However, the whole multi-parameter analysis established here aims at minimising the detection of low altitude transported dust (and thus the false positive detection of dust emission events) by using surface state parameters and 10m wind speed. Therefore, if dust is found in the lowest atmospheric layer at a place that permits dust emission due to the wind speed and soil surface criteria, we should consider that this is a plausible local emission. In essence, this is the whole basis of our analysis. Now, obviously, this does not mean that the local emission event is responsible for all the observed dust, and dust emission can certainly occur along with low altitude transport if the conditions allowing for dust emission persist (as mentioned in section 5 when comparing with the SEVIRI-based manual tracking). Furthermore, the analysis presented here does not intend to quantify the amount of dust emissions linked to each event (or in total), but to map these areas prone to dust emissions, even if only at specific times of the day. Therefore, it is difficult to compare with, for example, Caton Harrison et al. (2019), where CPO is reported to be responsible for 82% of total observed dust (in central and West Sahara during summer). Our analysis counts the number of events, while Caton Harrison et al. (2019) reports mechanism attribution linked to emitted dust amounts. In a nutshell, Caton Harrison et al. aims at attributing the primary meteorological driver, whereas this study aims at mapping the active dust source regions.

**2. I am not aware of any evidence that ERA5 is capable of representing winds associated with CPO activity. Could this algorithm risk unfairly excluding many of those events with high CPO winds that are severely underrepresented in ERA5?**

Indeed this under-representation of high wind speed in wind models is known, although to our knowledge no evaluation of the ERA-5 data set regarding the representation of deep moist convection over North Africa has been published so far. We assume that the model performance is not worse than the performance of the ERA-Interim data set. We do mention this (but rather briefly) in section 3.3.1: "We selected a lower threshold wind velocity of 5 m/s to possibly foster dust emission. This threshold is on the lower side (Marticoréna, 2014;

Marsham et al., 2013; Kocha et al., 2013), to cope with the well-known underestimation of high wind speed in models (e.g. Largeron et al., 2015)."

So we try to take this possible underestimation into account by using the lowest wind speed threshold in the analysis. The sensitivity to that threshold was analysed with regards to the comment made by referee #2 (comment #8). Please see our reply above.

**3. For the introduction: please consider discussing more the challenges and opportunities associated with satellite dust source detection/analysis? Maybe less space could be devoted e.g. to the microscale details of dust emission to accommodate this. I feel it would be beneficial to explicitly justify the current research in the context of the literature, other than just providing 'still another approach'. How has dust source detection been accomplished and what exactly is missing that this work addresses? If it's the vertical component, please also compare with previous attempts at this, e.g. Todd and Cavazos-Guerra (2016). See also Bakker et al. (2019) for a recent dust source analysis in ACP. A preprint article by Chédin et al. (2019) on IASI dust observations (https://www.essoar.org/doi/10.1002/essoar.10501119.1) may also be of interest.**

Many thanks for these comments. We've considered your suggestion(s) carefully. We now further refer to the study by Todd and Cavazos-Guerra (2016) in the introduction. The analysis based on CALIOP has the advantage of high resolution vertical profiles but the limitation of extremely poor ground coverage (very thin track every 1200km in the Tropics), missing many events. The following sentence was added (line 94):

> Todd and Cavazos-Guerra (2016) used CALIOP aerosol vertical profiles to calculate a dust emission index for the Sahara during the summer.

The Bakker et al. (2019) work, although extremely interesting, is not mentioned in our manuscript as their goal is not to identify source areas via their emission activity, but to identify the geomorphology of some major source areas during a specific period. They used SEVIRI dust RGB colour composite to visually (manually) identify dust plumes, tracked the 20 largest plumes to their sources and then used Sentinel-2 imagery to characterise those areas with high geographic resolution. They also quantified the emitted dust mass per source region.

Regarding the introduction discussing more challenges and opportunities, please also see the answer to referee # 2's comment 1.

Many thanks as well for making us aware of the submitted manuscript by Chédin et al. It was submitted about 3 months before we submitted our manuscript, so both analyses were run in parallel. Both analyses based on IASI dust data are very different. It would be good to compare them in the future, however, this is beyond the scope of this manuscript.

**4. Surface and soil moisture thresholds (algorithm steps 5 and 6): is there any way you could give a sense of what proportion of pixels which would have otherwise passed were filtered out by this step? It looks from Figure 4 like mainly the Sahel region will be affected. As you identify later this is a crucial factor as our understanding of Sahel dust sources is limited.**

This soil moisture filter is used only in areas that are not always bare and may be - at least partly - vegetated at some time throughout the course of the year, such as the blue parts in Figure 4. Indeed it is mostly affecting the (southern) Sahel. Figure 1 here shows the relative difference of the complete multi-parameter analysis (as in the manuscript figure 6) and the analysis without the soil moisture and NDVI filters. One can see that the large majority of the events are discarded as plausible local emissions in the southern part of Sahel due to the constraints on soil moisture and vegetation during fall and beginning of the winter while a large majority of events during spring are considered as plausible local emissions.

**5. Figure 5: The Tademait Plateau itself seems quite unlikely to be a dust source, compared to the large array of sebkhas in the Tidihelt basin directly south (e.g. see Ashpole and Washington, 2013) – please double check whether the plateau is actually collocated with your results?**

Indeed, the emission areas highlighted in our work are surrounding the Plateau but not the Plateau itself. The caption of Figure 5 was not modified as it lists the geographic features and not the dust sources. The only text

[Figure]

Figure 1: Relative difference of the complete multi-parameter analysis (as in the manuscript figure 6) and the analysis without the soil moisture and NDVI filter.

part where the Tademaït is specifically mentioned is in section 5.1, line 378. We added "around" before "the Tademaït Plateau".

**Technical corrections**

Changes done in the text are given as answer to each comment: the original text in normal characters and the changes either as crossed out removed text or added bold text.

**A few small grammatical/language issues need addressing but don't affect general readability. Some examples are "estimate on" rather than "estimate of", sentence starting "The Angstrom exponent. . ." on L124, L191 "that data set" instead of "this data set", L308 "All area". . .One repeated error is the use of "allows to" or "allowing to".**

"Estimate on" only occurred line 118 and was modified.

L124: We modified the sentence:

> The Angström exponent is a proxy for the particle size **and the particle size**  distribution changes with residence time and transport distance**:**  large particles drop out more quickly (Allen et al., 2013).

L191: done.

L308: The sentence was modified:

> All area highlighted here potentially also include a downwind dust transport component. **This component is linked either**  to the transport between sunrise at about 6:00 LST and the measurement time at about 9:30 LST, or **to the fact that**  end afternoon convection events **are**  detected a bit later**,** at 21:30 LST.

Allows to / allowing to :

- line 14:

  > The findings of our study illustrate the spatio-temporal distribution of North African dust sources based on 9 years of data, allowing  **for the observation of** a full seasonal cycle of dust emissions [...]

- line 116: dealt with in the specific comment below

- line 320:

  In the Sahel, the additional criteria linked to surface conditions are extremely meaningful, allowing **one** to account for the seasonal cycles of vegetation and humidity in addition to the significant changes in winds.

- line 431:

  Our data provide for the first time a monthly aggregation over 9 years, allowing  **for the extension of** the source analysis to a long period, with a full seasonal pattern.

- line 445:

  (3) the automated processing allowing  **for a quick analysis of** large areas for multiple years.

- line 459:

  In the long term, such an analysis will be possible with longer time series, allowing  **for the** study **of** possible trends in dust emissions.

**L25: "Saharan"**

Saharan **desert**

**L90: The Ashpole and Washington scheme temporal sample is 2004-2010, which is long but only applied for summer, and the geographical area is the western Sahara which is not complete but does not seem small?**

Ashpole and Washington (2013) have developed a way of automatically tracking dust plumes observed by SEVIRI to identify their source regions  **and applied it to the summertime western Sahara**.

**L103: "This eventually. . ." sentence unclear, please rephrase**

 **At the end of the process we obtain** a selection of low-altitude dust plumes which geo-location can be considered as active dust source.

**L105 ", and," remove first comma**

Done

**L116 "allows to gain" rephrase**

Using IASI therefore  **provides** information at times right after the two diurnal emission peaks.

**L118: "estimate on" −> "estimate of"**

Done

**L128 "by contrast" more appropriate than "on the contrary"**

Done

**L134 "provide very good prerequisites" unclear – perhaps a "good basis"?**

Done

**L135 Minor but the first point is stated here for the first time whereas 'in a nutshell' implies it isn't.**

Indeed. As your next comment recommends to move the paragraph mentioning the long time series, this is now correct.

**Paragraph 1 of section 2.1 would make more sense at the start of section 2**

Agreed and done.

**L187 "scares" − > "scars"**

Done

**L200 "a hint to. . ." I'm not sure what this means. Perhaps you mean that the corresponding figure is referenced in the schematics?**

Indeed, we mean that the schematics contains also the figure numbers matching specific steps. We rephrased, hoping it is more clear:

>  **The figure numbers illustrating some steps are** also given in the schematics.

**Figure 2: a discrete colour bar would be much easier to read than a continuous one. The same applies to all other map figures.**

The continuous colour bar allows for reflecting the spatial heterogeneity in more detail than a discrete colour bar would. It further avoids creating arbitrary categories which may bias the reader.

**L236 Not clear what "those" refers to**

We meant the ones discussed in the line above.

> Our selected near-surface layer concentration threshold is a stronger constraint than  **the above-mentioned** total column  thresholds, but ensure that those are real events.

**Section 4.1 please refer to the label numbers in Fig 5 as you discuss geographical locations.**

Agreed and done using e.g. "(Brown 1 in Fig. 5)". A sentence was added before section 4.1:

> **The numbers and colours in that figure are used to reference the geographical areas within the text.**

**Section 5.1 please continue referring to label numbers in Fig 5 where possible.**

Done, same as previous comment.

**L402 should be "two pieces of evidence"**

Done

**L451 "emissions do. . ." unclear, please rephrase**

> In that case, probably this new method overestimates the number of events, although it may be argued that if conditions remain stable, emissions  continue along time ans along a significant distance and should be counted multiple times

---

## Author Response (AR2)

**Author comment on "North African mineral dust sources: new insights from a combined analysis based on 3D dust aerosols distributions, surface winds and ancillary soil parameters"**

Sophie Vandenbussche      Sieglinde Callewaert      Kerstin Schepanski

Martine De Mazière

September 10, 2020

We thank the 2 referees and the editor for the final revision and comment on our manuscript. Here under is the final comment from anonymous referee 2 (in bold) and our answer (in plain text).

**Comment1: It's unnecessary to include Table 1 in AC1-supplement in the revision. Briefly introduce the difference between this one and other published methods would be sufficient to demonstrate the innovation of this study.**

We have removed the table and the line referring to it from the manuscript. The current paragraph starting line 100 was rewritten during the major revision and we think that it does indeed highlight the innovation of our study.

[revised manuscript text omitted]